# Comparative Study of *Bacillus*-Based Plant Biofertilizers: A Proposed Index

**DOI:** 10.3390/biology13090668

**Published:** 2024-08-28

**Authors:** Adoración Barros-Rodríguez, Pamela Pacheco, María Peñas-Corte, Antonio J. Fernández-González, José F. Cobo-Díaz, Yasmira Enrique-Cruz, Maximino Manzanera

**Affiliations:** 1Institute for Water Research and Department of Microbiology, University of Granada, 18071 Granada, Spain; dorybarros12@gmail.com (A.B.-R.); pamelapachecod@gmail.com (P.P.); 2VitaNtech Biotechnology S.L, Av. de la Innovación, 1, 18016 Granada, Spain; info@vitantechbio.com; 3Biopharma Research S.A (ECONATUR Group), P. Industrial Autovía Norte, C/ Montecillo S/N, 14100 La Carlota, Spain; 4Estación Experimental del Zaidín, Department of Soil and Plant Microbiology, Consejo Superior de Investigaciones Científicas, 18008 Granada, Spain; antonio.fernandez@eez.csic.es; 5Department of Food Hygiene and Technology, Universidad de León, 24071 León, Spain; jcobd@unileon.es

**Keywords:** biofertilizer efficiency, microbial biodiversity, biofungicides, agricultural protection against stress index

## Abstract

**Simple Summary:**

Comparing different commercial biofertilizer products for their effects as biostimulants, as protectors of plants from drought and plant-pathogenic fungi, can be difficult due to the increasing number of bacteria available in the market. Using a panel of tests, we compared the effects of four commercial strains of the operational group *Bacillus amyloliquefaciens* to assess their ability to protect pepper plants and stimulate their growth. Then, we obtained numerical values that allow quick diagnoses when choosing a product to ensure that it has the greatest added value or when describing a new strain.

**Abstract:**

The market for bacteria as agricultural biofertilizers is growing rapidly, offering plant-growth stimulants; biofungicides; and, more recently, protectors against extreme environmental factors, such as drought. This abundance makes it challenging for the end user to decide on the product to use. In this work, we describe the isolation of a strain of *Bacillus velezensis* (belonging to the operational group *Bacillus amyloliquefaciens*) for use as a plant-growth-promoting rhizobacterium, a biofungicide, and a protector against drought. To compare its effectiveness with other commercial strains of the same operational group, *Bacillus amyloliquefaciens*, we analyzed its ability to promote the growth of pepper plants and protect them against drought, as well as its fungicidal activity through antibiosis and antagonism tests, its ability to solubilize potassium and phosphates, and its ability to produce siderophores. Finally, we used a probit function, a type of regression analysis used to model the outcomes of analyses, to quantify the biostimulatory effectiveness of the different plant-growth-promoting rhizobacteria, developing what we have called the Agricultural Protection Against Stress Index, which allowed us to numerically compare the four commercial strains of the operational group *Bacillus amyloliquefaciens*, based on a Delphi method—a type of regression analysis that can be used to model a cumulative normal distribution—and integrate the results from our panel of tests into a single value.

## 1. Introduction

Agriculture is one of the human activities that has most significantly impacted the biodiversity of terrestrial ecosystems. From its early practices, such as flooding soils and burning forests to prepare land, to more recent methods involving extensive chemical use—fertilizers, herbicides, and pesticides—soil biodiversity has been continuously affected [1]. Agricultural practices consume approximately 70% of the world’s fresh water, necessitating reservoir construction and river diversion, which not only contaminate water sources but also lead to issues like eutrophication, adversely affecting the quality of both aquatic ecosystems and terrestrial biodiversity [2].

In response to these challenges, alternative cultivation techniques are being proposed. These involve environmentally friendly irrigation systems, soil conservation methods, and the use of biostimulants and biofertilizers—consisting of microorganisms—that promote plant growth, produce fungicides, and reduce water consumption [3]. Recent reviews highlight the significance of these approaches in mitigating the detrimental impact of conventional agricultural practices [4].

Furthermore, the new European Common Agrarian Policy (CAP) aims to achieve a 50% reduction in the use of high-risk phytosanitary products (i.e., insecticides, fungicides, herbicides, rodenticides, and nematicides of chemical origin) and a 50% decrease in soil nutrient losses by 2030. Notably, this policy seeks to limit the use of conventional fertilizers by 20% and promote the increased adoption of biostimulants and biofertilizers as essential alternatives. An additional factor driving these changes is the rising consumer demand for organic foods, which has surpassed EUR 106 billion globally. Within the European market alone, this expenditure is EUR 46 billion, with a future projection that organic production will occupy 25% of available farmland [5].

Biofertilizers, which comprise viable algae, fungi, and bacteria, serve to support plant growth and enhance crop yields [6]. They enrich soil organic matter, make nutrients accessible through solubilization, and contribute to nitrogen fixation in leguminous crops and the production of plant hormones that mitigate biotic and abiotic stresses [7]. The efficacy of microbial consortia rests on the interactions among individual strains, which exhibit various phenotypic relationships, such as the secretion of acids and secondary metabolites and the formation of biofilms that facilitate or hinder the proliferation of other microbes [8].

Historically, the understanding of bacterial antagonism—where bacteria inhibit the growth of other pathogens—has been recognized since the early 20th century. Current research shows that major bacterial phyla possess antagonistic pathways, with substances involved in these processes including antibiotics and protein toxins [9].

Biofertilizers can be categorized into different types, ranging from nitrogen-fixing strains to those that solubilize phosphates, potassium ions, and other micronutrients [10]. Nitrogen-fixing biofertilizers, such as *Azotobacter*, *Nostoc*, and *Rhizobium*, enhance soil nitrogen levels by fixing atmospheric nitrogen [11]. Meanwhile, phosphate-solubilizing and potassium-mobilizing biofertilizers include microorganisms like *Bacillus* and *Pseudomonas* [12].

Given the diverse range of bacteria that function as plant-growth-promoting rhizobacteria (PGPR), a standardized system for comparing these microorganisms is necessary for the benefit of end users. This study focused on isolating a *Bacillus* strain beneficial for promoting plant growth, controlling phytopathogenic fungi, and providing drought protection. We propose a comparative system to evaluate the efficacy of this strain against other commercially available strains, such as *Bacillus amyloliquefaciens* DSM7 [13,14], D747 [15], FZB42 [16], and DSM7 [17]. Each of these strains is noted for its biofungicidal activity, enhancing plant growth via indole acetic acid (IAA) production, among other mechanisms. For instance, *B. amyloliquefaciens* D747, marketed under different names, including Valcure, Revitalize, Amylo-X WG, and Garden Friendly Fungicide, is recognized for its ability to induce systemic resistance and increase flowering in crops like saffron, while *B. amyloliquefaciens* FZB42, available through Syngenta under the name Taegro, as well as from Abitep under the trade name RhizoVital 42 [17], is noted for its protective and curative biofungicidal properties [18]. And *B. amyloliquefaciens* DSM7 ATCC 23350 Fukumoto, marketed by Sinobest, produces biofungicides and has plant-growth-stimulating effects thanks to, among other factors, its ability to produce indole acetic acid (IAA) [19].

This study aimed to provide a quantitative comparison of these *Bacillus* strains’ abilities to promote plant growth, protect against drought, and combat phytopathogenic fungi.

## 2. Materials and Methods

### 2.1. Microorganisms, Media, and Culture Conditions

The type strains of *Bacillus amyloliquefaciens* D747, *Bacillus amyloliquefaciens* subsp. *amyloliquefaciens* Fukumoto DSM7, and *Bacillus velezensis* FZB42 were obtained from Deutsche Sammlung Von Mikroorganismen und Zellkulturen GmbH, Braunschweig, Germany. The bacterial strains used in this study are shown in Table 1. The bacteria were grown in tryptic soy broth (TSB) at 30 °C and 180 rpm, and solid media agar (1.5% *w*/*v*) was used to make tryptic soy agar (TSA) [20].

### 2.2. Isolation of Drought-Protecting Bacterial Strains

Soil samples (beige to brown clay–loam; moderate, medium granular structure), taken from the proximity of the roots of lettuce plants (*Lactuca sativa* var. *capitata*), were collected from a rainfed area subject to seasonal drought in Las Gabias, Granada, Spain (37°10′55″ N, 3°41′20″ W), according to the method developed by our research group [20]. To isolate spore-forming microorganisms, 1 g quantities of rhizospheric soil samples were air-dried and treated with pure chloroform (99.8%) for 60 min with occasional vortexing, as described in [4,20], since this process is detrimental to most vegetative cells, while many spores resist treatment with organic solvents. After complete evaporation of the chloroform, the soil samples were added to 10 mL TSB and thoroughly mixed. After the soil particles settled, serial dilutions in saline solution (0.9% *w*/*v* sodium chloride) were made and 100 μL aliquots from each dilution were plated on tryptic soy agar (TSA) plates [23]. After 48 h of incubation at 30 °C, single colonies were selected from the agar plates and re-streaked onto fresh selective media to obtain pure cultures. To select for different isolates, morphological (size, shape, color, and texture of colony) and growth characteristics (rate of growth, colony edge, and elevation) were examined. This step was repeated at least three times to ensure the elimination of contaminants. To confirm the purity of the cultures, individual colonies from each re-streaked plate were subjected to microscopic examination. A sterile slide and coverslip were used to observe colony morphological and cellular characteristics under a light microscope. The absence of contaminants was verified by assessing the homogeneity of cell morphology. When required, molecular identification was performed by DNA extraction from purified cultures using the DNeasy^®^ Blood and Tissue Kit (Qiagen, Venlo, The Netherlands), following the manufacturer’s instructions. The quantity and quality of the extracted DNA was determined using a NanoDrop 2000 (Thermo Scientific, Wilmington, CA, USA). Polymerase chain reaction (PCR) was performed with specific primers targeting the 16S rRNA gene to amplify a region characteristic of the genus and species, as previously described [23]. PCR products were analyzed via agarose gel electrophoresis and sequenced as described below. After the sporulation test, an assay on their ability to protect plants from drought was performed. To that end, pepper plants (*Capsicum annuum* L. *cv*. Maor) were inoculated as described below and kept without watering. Microorganisms with the ability to protect pepper plants that presented a relative water content higher than that of *Microbacterium koreensis* 3J1 were considered drought-protectant.

### 2.3. Sporulation Test

To determine if the isolated strains were able to produce spores, a 3-day-old colony taken from TSA plates incubated at 30 °C, after being re-streaked, was resuspended in 1 mL of sterile saline solution and incubated for 30 min at 72 °C, as previously described [24]. Aliquot volumes (10 μL) were plated on TSA plates before and after the 72 °C incubation and incubated again at 30 °C for 3 days. Strains able to grow on TSA plates at 30 °C under both conditions were selected as spore-producing or thermotolerant bacteria. Tests were performed in triplicate.

### 2.4. Optical and Electronic Microscopy

Prior to electronic microscopy, optical microscopy was used to ensure the presence of spores. For spore staining, 100 mL flasks containing 10 mL of TSB were inoculated with the different candidates and incubated at 30 °C with shaking at 180 rpm (Infors HT Multitron, Bottmingen, Switzerland). After 96 and 120 h of incubation, spore staining was performed by fixing the samples on slides, adding malachite green (5% *w*/*v*) in vapor emission, and using 1% (*w*/*v*) safranin as a contrast dye. Once the presence of spores was confirmed by optical microscopy, 1500 µL fractions of the cultures were placed in microtubes. Finally, the samples were taken to the Scientific Instrumentation Center of the University of Granada (CIC-UGR), where they were prepared independently for observation via scanning and transmission electron microscopy (SEM and TEM).

For sample preparation, a 100 mL flask containing 10 mL of TSB broth was inoculated with the *Bacillus* sp. A6 strain and incubated for 120 h at 30 °C with shaking at 180 rpm. Samples (5 mL) were pelleted at 14,000 rpm for 5 min, washed with 1 mL of phosphate-buffered saline (PBS), and finally fixed in 1 mL of 4% paraformaldehyde and 2.5% glutaraldehyde in cacodylate buffer (0.05 M, pH 7.2). Samples were kept refrigerated for up to 24 h. Then, the samples were washed three times with cacodylate buffer (0.1 M, pH 7.2) and incubated at 4 °C for 30 min. Finally, the samples were resuspended in cacodylate buffer and processed by the CIC-UGR. Sample preparation for SEM was performed as previously described [25,26,27]. To that end, cells were located on a surface with 0.1% poly-*l*-lysine and incubated for fixation for 20 h at 4 °C in an atmosphere saturated with glutaraldehyde. Then, 1 h incubation was performed with 1% osmium tetroxide. Finally, samples were critical-point dehydrated [27] using carbon dioxide and coated with carbon (EMITECH K975X, Quorum Technologies Ltd., East Grinstead, UK). For TEM, samples were treated following the method described in [28]. In summary, the samples were fixed with 1% osmium tetroxide and 1% potassium ferricyanide at 4 °C for 1 h. Then, EMbed 812 resin was used to embed the samples. A DIATOME diamond knife (Ultracut R. LEICA, Wetzlar, Germany) was used to generate ultrathin sections (50–70 nm) from the polymerized resin blocks. Sections were mounted on 200-mesh copper grids. The grids were post-stained with 1% uranyl acetate and subsequently with lead citrate, as previously described [29]. The samples were observed using transmission electron microscopy (Carl Zeiss SMT LIBRA 120 PLUS Transmission Electron Microscope, Oberkochen, Germany) and scanning electron microscopy (CARL ZEISS GEMINI High Resolution Field Emission Scanning Electron Microscope FESEM, Oberkochen, Germany) at the Microscopy Service of the CIC-UGR.

### 2.5. Monitoring of Plant Growth

For Italian sweet pepper plants (*Capsicum annuum* L. *cv*. Maor), the dry weight (DW), fresh weight (FW), and fully turgid weight (FTW) of the whole plants free from soil were measured four times after inoculation at 0 and 31 days after inoculation and expressed in grams. The relative water content (RWC) was calculated according to [4,30] as follows: RWC = (FW − DW) × (TW − DW)^−1^. The relative water content is a dimensionless parameter whose maximum theoretical value is 1 in a state of maximum hydration. In addition, root length (RL) and stem length (SL) were recorded as described elsewhere [30]. To ensure consistency in measurements across different time points to determine fresh weight, plants were harvested at the same time of day (10:00 am) to avoid variations due to transpiration. A precision scale was used to record weight immediately after harvest. To obtain the fully turgid weight, the samples were immersed in distilled water for 48 h in the dark. After this period, they were removed, gently dried with absorbent paper, and weighed again. As for the dry-weight measurement, the samples were placed in an oven at 70 °C for 72 h to ensure complete removal of moisture. Once cooled in a dehydrator, they were weighed to obtain the dry weight.

### 2.6. Plant-Growth Conditions, Bacterial Inoculation, and Plant Sampling

One month-old Italian sweet pepper plants (*Capsicum annuum* L. *cv*. Maor) were purchased from the plant nursery SaliPlant S.L. (Granada, Spain). The pepper plants were incubated in a growth room at 50–60% constant relative humidity. The temperature of the room was maintained at 20 °C and lit with a 12 h day/night cycle and gradual dimming/brightening of the light to simulate dawn and dusk. The day cycle consisted of 200 μmol photons·m^−2^·s^−1^, and the dawn–dusk cycle consisted of 150 μmol photons·m^−2^·s^−1^. The plants were regularly watered, unless otherwise indicated. For the inoculation with the different strains of inocula, and to guarantee a similar number of bacterial cells in each plant and that their effect came from the plant–microorganism interaction, without taking into account the presence of secondary metabolites produced throughout the culture, the cells were separated from the culture by centrifugation and resuspended in M9 buffer. These inocula were supplied by VitaNtech Biotechnology, Granada, Spain, and the plants were treated with 3 mL of the liquid inoculant (consisting of a bacterial suspension from the enriched culture on M9 buffer (Na_2_HPO_4_·7H_2_O, KH_2_PO_4_, NH_4_Cl, and NaCl) at an absorbance of 1 at a wavelength of 600 nm), representing a concentration between 1·10^6^ and 1·10^8^ colony-forming units (CFUs). The time was recorded as time 0 of the assay, and no additional water was supplied during the experiment for the drought tests, as described previously [31].

### 2.7. Phosphate Solubilization Test

The phosphate solubilization test was carried out using SMRS1 medium, which contained Ca_3_(PO_4_)_2_ as a source of inorganic phosphate, along with (NH_4_)_2_SO_4_, KCl, MgSO_4_, MnSO_4_∙H_2_O, FeSO_4_∙7H_2_O, NaCl, glucose, yeast extract, and bromocresol purple as a pH indicator. Then, 20 µL of each strain suspended in 0.85% saline solution was inoculated on SMRS1 agar plates. After 24 h of incubation at 30 °C, the phosphate solubilization capacity was checked by the presence of a yellow halo around the inoculum due to the acidification of the environment [32]. The diameters of the halos were measured, and *Pseudomonas putida* KT2440 was used as a positive control and *Arthrobacter* sp. 5J12A as a negative control.

### 2.8. Potassium Solubilization Assay

To find out if a strain was able to solubilize potassium salts, we used culture medium containing glucose, MgSO_4_∙7H_2_O, FeCl_3_, CaCO_3_, Ca_3_(PO_4_)_2_, KCl, K_2_SO_4_, Phenol Red, and agar. The strains were reactivated on TSA agar plates and subsequently streaked in the medium described above to evaluate their ability to solubilize potassium salts. They were incubated at 30 °C for 7 days. The strains’ ability to solubilize potassium was determined through the presence of yellow halos around the colonies. *P. putida* KT2440 was used as a positive control, and *Escherichia coli* OP50 was used as a negative control.

### 2.9. Siderophore Production Assay

For siderophore production, we used a test based on CAS medium, which consisted of “CAS” solution containing CAS (Chromium blueS); FeCl_3_ dissolved in HCl and HDTMA (Hexadecyltrimethylammonium bromide); the minimal medium M9, which contained KH_2_PO_4_, NaCl, and NH_4_Cl as components; glucose; NaOH pH 12; casamino acid solution; PIPES; and agar. The strains were streaked on TSA agar and allowed to grow for 24 h at 30 °C. After incubation, an overlayer of CAS agar tempered at 45 °C was added over the grown colonies, and, finally, the plates were incubated again at 30 °C for 12 days. The production of siderophores by the strains was determined by the color change from blue to yellow in the CAS medium [33]. *P. putida* KT2440 was used as a positive control, and *Proteus* sp. S47 [22] was used as a negative control.

### 2.10. Quantification of Phytohormones: Indole Acetic Acid, Indole Butyric Acid, and Gibberellins

For the quantification of plant hormones produced by the bacterial strains, the latter were inoculated in 20 mL of TSB medium and incubated at 30 °C, with shaking at 180 rpm for 24 and 72 h. Phytohormones were extracted from the culture supernatants by acidifying them with 2 N HCl until a pH of 2.5 was reached. Then, the same volume of ethyl acetate was added, followed by evaporation of the ethyl acetate fraction. Finally, the remaining fraction was dissolved in 1 mL of HPLC-grade methanol. The calibration curve was prepared using methanolic solutions of the standards corresponding to each phytohormone: 3-Indolebutyric Acid (TCI America), 3-Indoleacetic acid (Sigma-Aldrich), and gibberellic acid A3 (Duchefa Biochemie). Concentrations of 1, 4, 20, 100, and 500 µg/L of each chemical were used. Finally, all samples, controls, and standards were injected into a WATERS model XEVO TQ-S mass spectrometer with a triple-quad analyzer at the Scientific Instrumentation Center of the University of Granada. In addition, two controls, *P. putida* KT2440 as a positive control and *E. coli* OP50 as a negative control, were included [34].

### 2.11. Determination Urea Hydrolysis

To determine their urease activity, the tested strains were sown on Christensen urea agar, which contains gelatin peptone, dextrose, NaCl, KH_2_PO_4_, phenol red, agar, and 2% urea. The color change of the medium from yellow to red shows the hydrolysis of urea. *Proteus* sp. S53 was used as a positive control, and *E. coli* OP50 was used as a negative control [35].

### 2.12. Antagonism and Antibiosis Assays

The antifungal tests were carried out based on two techniques known as antagonism and antibiosis, which we describe below using *Botrytis cinerea* and *Fusarium oxysporum* as target pathogenic fungi.

Antagonism

To analyze the antagonistic effect on the fungi, they were grown on PDA plates for a week at 30 °C before the addition of the bacteria. The addition of the bacteria was considered time zero. The bacterial inocula were cultured 24 h prior to the addition in test tubes with 5 mL of TSB at 30 °C and 180 rpm. The PDA plates were inoculated with 100 μL of bacterial culture. Once the PDA plates were inoculated with the bacterial culture, a piece of fungus one centimeter in diameter was added to the center of each PDA plate, and they were incubated in a thermostat at 28 °C until the completion of the experiment [36].

Antibiosis

To analyze the effect of the bacterial strains on the fungi, the fungi were previously grown for a week at 30 °C. A piece of the fungus measuring one centimeter in diameter was added to one side of each PDA plate, and this was considered time zero. Bacterial inocula were grown for 24 h on TSA plates in an oven at 30 °C four days before time zero. The PDA plates inoculated for a week with the fungus were inoculated with the bacteria; a colony isolated from the TSA plate was sown at one end of each PDA plate, three days after time zero, and incubated in a thermostat at 28 °C for the duration of the experiment [37].

### 2.13. Extraction of Nucleic Acids and Next-Generation Sequencing

Nucleic acid extraction, library preparation, and Illumina sequencing were carried out at the IPBLN Genomics Facility (CSIC, Granada, Spain), as previously described [23].

### 2.14. Genome Sequencing 

*Bacillus* strain A6 genomic DNA (gDNA) was extracted using the DNeasy^®^ Blood and Tissue Kit (Qiagen, Venlo, The Netherlands), following the manufacturer’s instructions. Quantification of the extracted DNA was performed using NanoDrop 2000 equipment (Thermo Scientific, Wilmington, CA, USA). The gDNA was sequenced via Illumina using a 150 bp PE strategy in the NovaSeq platform.

Raw sequences were quality-checked with FASTQC version 0.11.5 [38], and forward filtering was not needed. To assemble and annotate this genome, the KBase [39] web platform was used, as detailed below. Firstly, paired-end reads were separately uploaded to the KBase server and then used to de novo assemble the genome with the SPAdes [40] App v3.13.0, together with QUAST [41] and CheckM [42], to check the assembly quality. Furthermore, genome contigs were remapped with Bowtie2, and contig consensus sequences were manually checked using Geneious Prime v2021.1.1. Reads were assembled into contigs by SPAdes v3.13.0 [27], using 55, 75, and 97 as k-mer sizes. Secondly, genome annotation was performed with the RASTtk [43,44,45] App v1.073, and the protein domains were identified and annotated using PSI-BAST against COGs, CDD, SMART, PRK and HMMER against Pfam, TIGRFAM, and NCBIfam using the Domains App v1.0.10. Finally, metabolic functions were obtained from the KEGG and MetaCyc databases using the Model SEED App v2.0.

Average nucleotide identity (ANI) and digital DNA–DNA hybridization (dDDH) were utilized for taxon delineation at the subspecific level based on EZBioCloud, as previously described [46].

### 2.15. Statistical Analysis

For the statistical treatment of the data, RStudio i386 software, version 4.0.3 (PBC, Boston, MA, USA, 2011), was used. An evaluation of significant differences between treatments was carried out by applying a 95% confidence interval. The data were analyzed using an ANOVA (analysis of variance) model to determine whether there were statistically significant differences between the means of the compared groups. If significant differences were found, a post hoc analysis was conducted using Tukey’s HSD (honestly significant difference) procedure to compare the means of all groups exhaustively, allowing for pairwise comparisons.

To assess the normality of the data, a Shapiro–Wilk test was performed, and it was considered that the data were normally distributed if the resulting *p*-value was greater than 0.05. Additionally, the Bonferroni outlier test was employed to identify potential outliers. This procedure was carried out by checking whether any observations deviated significantly from the evaluated normal distribution. If any significant deviation was noted, the corresponding data were excluded from the analysis to ensure the robustness of the results.

It was considered that the difference between treatments was statistically significant when the obtained *p*-value was less than 0.05, allowing us to reject the null hypothesis of the equality of means.

## 3. Results

### 3.1. Isolation of a Collection of Drought-Protecting Spore-Forming Strains

Bacterial cells with the ability to protect plants from drought were isolated as described in Section 2. Multiple strains were selected, and pure cultures were obtained, resulting in a total of 209 isolates.

To differentiate those microorganisms that formed spores from those incapable of spore formation, we performed a sporulation test based on the temperature tolerance of spores [24,47]. A total of 67 isolates out of the initial 209 isolates resulted in temperature-tolerant strains. Then, we tested the ability of the spore-forming isolates to protect pepper plants from drought, following the method described in Section 2. Samples were taken at 0 and 31 days after inoculation and cessation of irrigation. Of the 67 isolates, an isolate named A6 showed the best drought-protection results (Figure 1), resulting in relative water content values even higher than those of the positive control, *Microbacterium koreensis* 3J1 [48]. To confirm the ability to form spores, spore staining and transmission and scanning electron microscopy were used, using a culture of A6 collected at 96 and 120 h (Figure 2).

### 3.2. Taxonomic Affiliation and Genome Sequence of Bacillus sp. A6

The A6 spore-forming isolate was identified by nucleotide sequencing of the near-complete 16S rRNA gene (1430 bp) (acc. no. PQ099275), which showed that it belonged to the operational group *Bacillus amyloliquefaciens* (OG*Ba*) [49] and showed similarity to *Bacillus siamensis* (acc. no. AJVF01000043), *B. velezensis* (acc. no. AY603658), *B. subtilis* (acc. no. ABQL01000001), and *B. amyloliquefaciens* (acc. no. FN597644), with a similarity of 99.93%, 99.92%, 99.78%, and 99.70%, respectively. Due to the peculiarities of this operational group, we decided to sequence the entire genome of strain A6 in order to determine its taxonomic affiliation in a more rigorous way (acc. no. SAMN40153660).

*Bacillus* sp. A6 gDNA sequencing generated 13,209,766 sequence reads 150 bp in length (1981 Mbp), resulting in a theoretical average depth coverage of ×495.25, assuming a genome size of 4 Mpb. After the assembly, 22 contigs were obtained with an N50 of 607,087 bp, and the largest was 1,036,557 bp. Furthermore, the total length of the assembled genome was 4,013,767 bp, with a GC of 46.32%, and the smallest contig had a length of 511 bp. Finally, 4.110 coding sequences, 52 non-coding rRNA genes, 13 non-coding repeats, and 5 non-coding prophages were annotated with the RAST algorithm, obtaining 3183 distinct functions.

Three different rRNA 16S sequences were found (I, II, and III), the above-described rRNA 16S sequence coinciding with sequence I. When an *EzBioCloud* search was performed using sequence II, we observed the highest similarities with *B. tequilensis* (acc. no. AYTO01000043), *B. cabrialesis* (acc. no. MK462260), *B. inaquosorum* (acc. no. AMXN01000021), and *B. stercoris* (acc. no. MN536904), with a similarity of 99.55% in all four cases. On the other hand, when the analysis was performed using sequence III, the highest similarities were found with *B. aerius* (acc. no. AJ831843), *B. altitudinis* (acc. no. ASJC01000029), and *B. xiamensis* (acc. no. AMSH01000114), with similarities of 97.37%, 97.30%, and 97.22%, respectively.

We used a phylogenetic tree based on digital DNA–DNA (dDDH) hybridization for calculating intergenomic distances, which were then converted to dDDH values. The pairwise dDDH estimates of the 36 most closely related genome sequences from the OG*Ba* strains are represented in Table 2. The evolutionary history was deduced using the maximum likelihood method based on Kimura’s two-parameter model. The tree with the highest log likelihood (−1149.7217) is shown. The percentage of trees in which the associated taxa were clustered is shown next to the branches. The initial trees for the heuristic search were obtained by applying the neighbor-joining method to a matrix of distances between pairs estimated by the maximum composite likelihood (MCL) method. A discrete gamma distribution was used to model evolutionary rate differences between sites (five categories (+G. parameter = 4.7378)). The tree is drawn to scale, with branch lengths measured as the number of substitutions per site. The analysis included 36 nucleotide sequences. The codon positions included were first + second + third + non-coding. In the final data set, there was a total of 142 positions, showing that *B. velezensis* CR-502^T^ was the closest phylogenetic species to *Bacillus* sp. A6 (Figure 3).

### 3.3. Plant-Growth-Promoting Rizhobacteria Traits of B. velezensis A6 and Their Closest Commercial Strains

Since members of the OG*Ba* are known as plant-growth-promoting bacteria (PGPB) due to their abilities to fix nitrogen, solubilize phosphates, and produce siderophores and phytohormones, as well as antimicrobial compounds, three commercial strains of the OG*Ba* were included in the study for comparison, including the *B. amyloliquefaciens* DSM7 strain Fukumoto, *B. amyloliquefaciens* FZB42, and *B. amyloliquefaciens* D747. A tray with 31-day old pepper plants was purchased and inoculated with bacterial suspensions of the three commercial strains plus *B. velezensis* A6 separately, and the dry weights of the plants were measured at time 0 and 31 days after inoculation (Figure 4). Although plants inoculated with the four *Bacillus* strains showed a higher dry weight than the control in the absence of microorganisms, in the case of plants inoculated with DSM7, the differences were not significant. On the other hand, the strains FZB42 and D747 induced plant growth to a significantly higher extent than that observed in non-inoculated plants—to a level similar to that shown by plants inoculated with the PGPR *P. putida* KT2440—while plants inoculated with the A6 strain showed the highest values for stem and root length and dry weight. Strains that produced a combined increase in root and stem size greater than the control without bacteria under irrigated conditions were considered PGPR, while strains that produced an increase in dry weight as well as relative water content in the absence of irrigation, as observed in non-inoculated plants, were considered protective against drought.

### 3.4. Drought Protection and Plant-Growth Promotion of Pepper Plants 

The abilities of the four strains of *Bacillus* used as agricultural amendments to protect Italian sweet pepper plants were compared. *B. velezensis* A6 showed the highest values for fresh weight, dry weight, and relative water content at all three sampling times. The abilities of strains DSM7, FZB42, and D747 to protect pepper plants against drought were similar to each other and were not distinguished from that of the control, *P. putida* KT2440, which was unable to protect plants against drought, nor from that of the control in the absence of inoculum (Figure 5).

In order to compare the growth-promoting effects of the four *Bacillus* strains, the previous experiment was repeated, but irrigation was maintained to avoid water stress. Once again, greater fresh weight, dry weight, and fully turgid weight values were observed in the plants inoculated with *B. velezensis* A6—above those observed with the other inoculants (DSM7, FZB42, and D747, as well as *P. putida* KT2440). On this occasion, significant differences were observed between the weights of all the plants with respect to the non-inoculated control. Similarly, the root length was, again, greater when the plants were inoculated with *B. velezensis* A6; however, no significant differences in stem length or relative water content were observed, regardless of inoculation conditions (Figure 5).

### 3.5. Biofungicidal Activity of the Bacterial Strains

Strains of the OG*Ba* often produce antimicrobials, including antibiotics and fungicides, that protect plants from attack by certain pathogens. To compare the protective capacities of the four *Bacillus* strains under study, antagonism and antibiosis studies were carried out. These studies were carried out on the phytopathogenic fungi *Fusarium oxysporum* and *Botritis cinerea*. All the strains under study showed significant differences in their inhibition of the growth of both tested fungal strains, except in the case of the DSM7 strain at 15 days, where the growth of *F. oxysporum* was of the same extent as the control in the absence of bacterial inoculation. Although this strain showed some antifungal activity at 5 and 10 days, its effect was the lowest of those observed among the *Bacillus* strains. The antagonism study against *F. oxysporum* after five days of testing showed greater fungicidal activity when strain D747 was used than with the rest of the strains. The second strain with the highest antifungal activity was A6, which showed similar antifungal activity to D747 at 10 and 15 days. They were followed in antifungal efficiency by strain FZB42 and, finally, by DSM7, as we have indicated previously (Figure 6A,B). When testing on *B. cynerea*, all strains, except DSM7, showed the same remarkable degree of growth inhibition. Again, the DSM7 strain had the lowest inhibitory effect on the growth of *B. cynerea*, which was only apparent at 5 and 10 days. No significant differences from the control were observed in the absence of bacteria on day 15 (Figure 7A,B).

### 3.6. Phosphate and Potassium Salt Solubilization

Another advantage that certain PGPR offer is their ability to solubilize phosphorus and potassium salts. The ability to solubilize these salts is measured based on the diameter of the coloured halo generated by a pH indicator. The phosphate solubilization capacity of the four *Bacillus* strains was evaluated using the SMRS 1 medium. This medium contains bromocresol purple as a pH indicator, so the strains that have the capacity to solubilize phosphates form a yellow halo in the medium. The largest yellow diameter was formed by the DSM7 Fukumoto strain, covering a surface of 3.3 ± 0.15 cm, with no significant differences to that produced by the phosphate solubilizer *P. putida* KT2440. Strains FZB42, D747, and A6 showed similar values, with no significant differences, ranging from 1.1 to 1.6 cm (Figure 8).

Another advantage that certain PGPR offer is their ability to solubilize potassium salts, which are essential macronutrients that play an important role in the growth and development of plants. Potassium ions deficiency results in poor root development, slow growth, and lower yields due to small seeds [50]. In the strains evaluated, it was observed that all have the capacity to solubilize potassium salts.

### 3.7. Siderophore Production

The production of siderophores by soil bacteria makes it possible to capture iron from the soil in conditions of iron deficiency. The siderophores produced by PGPR help meet the iron needs of plants by causing its solubilization and chelation from organic or inorganic complexes present in the soil [51,52]. All the analyzed *Bacillus* strains showed the ability to produce siderophores.

### 3.8. Urea Hydrolysis

Urease carries out the hydrolysis of urea into ammonia and carbon dioxide, which is essential to supply plants with nitrogen [53]. However, none of the *Bacillus* strains tested were able to hydrolyze urea.

### 3.9. Quantification of Phytohormones: Indole Acetic Acid, Indole Butyric Acid, and Gibberellic Acid

Phytohormones produced by microorganisms are studied for their importance in promoting plant growth, as well as for their role in the protection of plants from stressors. We tested the ability of the *Bacillus* strains to produce IAA, one of the most abundant auxins, which plays a key role in the regulation of various physiological processes, such as cell division and elongation, vascular differentiation, gravitropism, and phototropism [54]. Among the tested strains, DSM7 was the largest producer of IAA with an even higher production of IAA (250 μg/L) than the positive control, *P. putida* KT2440 (212 μg/L), at 48 h. The rest of the strains showed very limited production of IAA in the range of 28–31 μg/L (Figure 9A).

Indole butyric acid (IBA) is another auxin produced by certain bacteria with an important role in plant growth. Similar to the IAA production, in the case of IBA production, DSM7 was the largest producer of this auxin (0.88 μg/L), followed by the other three strains, with values ranging from 0.5 (FZB42) to 0.64 μg/L (D747) (Figure 9B).

Gibberellic acid is also involved in different plant developmental processes and the regulation of many physiological processes [55]. Once more, it was the DSM7 strain that exhibited the largest production of the plant hormone. In this specific case, it was the only producer of gibberellic acid, with over 1.23 μg/L, which had a level even higher than that recorded for the positive control, *P. putida* KT2440 (0.87 μg/L). No production of gibberellic acid was detected for the other *Bacillus* strains (Figure 9C). In all cases, gibberellic acid was only found at 24 h and not at 72 h.

### 3.10. Agricultural Protection against Stress Index (APSI)

Although there have been numerous comparative studies of the effectiveness of certain plant-growth-promotion activities of different PGPR, to our knowledge, there have been no attempts to integrate the results of different experimental tests to derive a single index.

We used a probit function to quantify the biostimulation effectiveness of different PGPR [21].

We used the Delphi method to integrate the results from our panel of tests into a single value [21]. Our ultimate aim was to develop a range of values that indicate whether a candidate PGPR strain of the OG*Ba* is useful as a plant-growth promoter. Values below a certain cutoff score (50 ± 0.5) indicate the need for additional PGPR tests before the candidate can be considered suitable for use as a PGPR. We attempted to develop a simple but accurate, rigorous, and relevant set of tests to help decision makers evaluate the biofertilization effect of potential PGPR before approving the use of a candidate organism. We also aimed to determine whether the results of tests in field trials were consistent with the results of the rest of the tests included in the panel to minimize the number of field trials needed.

We termed the scale of values the Agricultural Protection Against Stress Index (APSI), which is scored from 0 to 100. Higher values indicate a greater likelihood that the bacterial strain of interest will be useful for use as a PGPR [21] (Table 3). The APSI is based on tests of growth stimulation of plants in the laboratory, protection against abiotic stress (drought), protection from biotic stress (antibiosis and antagonism against *F. oxysporum* and *B. cynerea*), production of plant hormones (IAA, IBA, and gibberellic acid), phosphate and potassium solubilization, and production of siderophores.

On the basis of earlier research, we assigned a given specific weight to each factor according to the following relative weights, where the ability to stimulate plant growth in the absence of stress represented 20%, the protection from water stress represented 27%, and the protection from phytopathogenic fungi represented 28% of the final APSI value. In addition, the ability to solubilize phosphorus or potassium salts represented 8%; the production of siderophores, 4%; and the ability to hydrolyze urea, an additional 4%. As shown in Table 3, the maximum possible sum of the values for the individual assays used to test plant-growth promotion, plant protection from biotic and abiotic stresses, the production of siderophores, urea hydrolysis, and solubilization of phosphate and potassium salts is 100; hence, the highest possible APSI score is 100.

To validate the potential applicability of this scoring system, we calculated the APSIs for four commercial strains of the *Bacillus* genus and contrasted the values with the results for their effects on lettuce plants obtained in a field trial.

As a result, we assigned an APSI score of 61.5 to *B. amyloliquefaciens* DSM7, a score of 54.5 to D747, a score of 62.25 to FZB42, and a score of 78.95 to *B. velezensis* A6, showing a clear correlation with the results from field trials on lettuces.

In a pilot field study carried out on maize under drought conditions, the effect of *B. velezensis* A6 on growth and productivity was evaluated by monitoring plant height, cob and grain weight, and yield, compared to *B. amyloliquefaciens* D747, among other biostimulant treatments. The results showed increases of 22.66% in height, 4.48% in corncob weight, and 12.20% in corn production measured in Kg/ha, compared to the results obtained with *B. amyloliquefaciens* D747 [56].

Pilot studies with twelve other PGPR candidates of the OG*Ba* yielded a similar degree of consistency between APSI scores and the results of field tests. To evaluate the potential of the APSI to reduce the number of field trials and be used to compare strains of PGPR of similar taxonomy, we compared these results in terms of weight, length, stress resistances, and plant hormone production.

## 4. Discussion

The use of bacteria as plants biofertilizers is increasingly recognized by farmers as end users. These biofertilizers can have different uses, ranging from plant-growth promoters to plant protectors against biotic and abiotic stresses [57]. The great diversity of products complicates decision making when purchasing a specific biostimulant [58]. In this work, we have developed an index to compare the effects of plant-growth-promoting bacteria and their abilities to protect plants from both phytopathogenic fungal attacks and water stress. We have called this index the Agricultural Protection Against Stress Index (APSI). The APSI allowed us to compare the efficiency of three commercial Bacillus strains belonging to the OGBa in a similar way to that in which other characteristics of other strains have been compared [21]. Furthermore, in this work, we isolated a soil bacterium belonging to the OGBa, which we characterized as *B. velezensis* A6, due to its ability to protect plants against water stress, as well as against attacks of *B. cynerea* and *F. oxysporum*. Although the DSM7 strain is quite weak in its ability to protect plants against drought, its score in the described index is compensated by its great capacity—greater than those of the rest of the strains under study—to solubilize phosphates and to produce plant hormones (AIA, IBA, and GB), as previously described [19]. By comparing the effects of the four strains in terms of various factors, such as the promotion of plant growth, protection against water stress, the production of siderophores, the ability to solubilize phosphates and potassium salts, as well as the ability to produce plants hormones, we managed to compare the four strains of the OG*Ba*. As a result of this comparison, we observed values from 61.5 to 78.95 points (out of the possible maximum of 100), and A6 stands out (with an APSI of 78.95) for its ability to protect plants from drought, while the strains FZB42, DSM7, and D747 failed to protect plants from drought. On the other hand, of these three strains, the D747 and FZB42 strains had very similar APSI values (65.25 and 67.75, respectively), which were higher than that of the DSM7 strain (with an APSI value of 61.5), as a result of their greater plant-growth stimulation in the laboratory, despite their lower production of phytohormones. This proximity in APSI values could be explained by their genetic proximity [59].

The advantage of the described index is its modular nature, whereby certain peculiarities are considered, such as the ability to protect plants against drought, stimulate their growth, and defend them from fungal attack. In environments devoid of these stresses, it may be more interesting to evaluate the ability to defend plants against other stresses, such as soil salinity or nematode attacks, so the APSI can be adapted according to the requirements of a particular environment [21,60].

To validate these results, we performed a small field trial based on iceberg lettuce subjected to water stress. The final weights of the plants showed a good correlation between the assigned APSI of the strain and the size and fresh weight of the different inoculated plants. Additionally, the APSI results were corroborated through an extensive corn field trial under drought conditions comparing plants inoculated with *B. velezensis* A6 with plants under the effect of *Bacillus amyloliquefaciens* D747. The APSI values for D747 reached 54.5, while those for A6 reached 63.7, which was reflected in the growth of the plant as well in the productivity in the field experiment using maize [56].

Additional studies with well-known plant-growth-promoting bacteria of the OG*Ba* are needed to validate the value of this index. Such additional assays with other bacteria could refine the model for future applications of the APSI and enable further quality assessments of this tool.

## 5. Conclusions

The newly isolated strain *B. velezensis* A6 is part of the OG*Ba*, with great potential to be used as a plant biostimulant for the protection of plants from hydric stress and from some phytopathogenic fungi. The newly described APSI allowed us to compare the efficacy of some well-known commercial strains, including FZB42, DSM7 (Fukumoto), and D747, and to assign numeric values to efficiently compare the four strains. Strain D747 stands out above the rest for its fungicidal capacity, although it does not show as powerful protection against drought as the other strains studied. The DSM7 strain stands out for its potential to solubilize phosphates and potassium salts, so its use may be recommended in soils with these types of salts, where cultivated plants do not have easy access to the ions derived from these salts.

## Figures and Tables

**Figure 1 biology-13-00668-f001:**
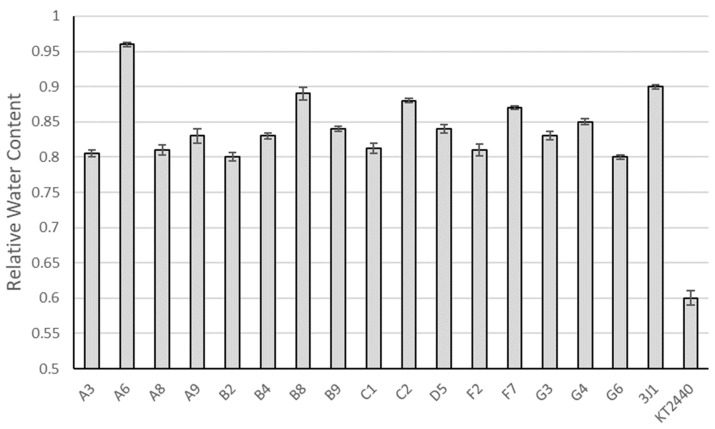
Relative water content of pepper plants subjected to drought and inoculated with different bacterial isolates. Relative water content of plants subjected to drought and inoculated with different isolates. The name indicated on the axis refers to the strain used. Only plants presenting values greater than 0.8 are displayed. Controls of plants inoculated with the drought-protectant *Microbacterium koreensis* 3J1 and the non-protective control, *Pseudomonas putida* KT2440, are also shown.

**Figure 2 biology-13-00668-f002:**
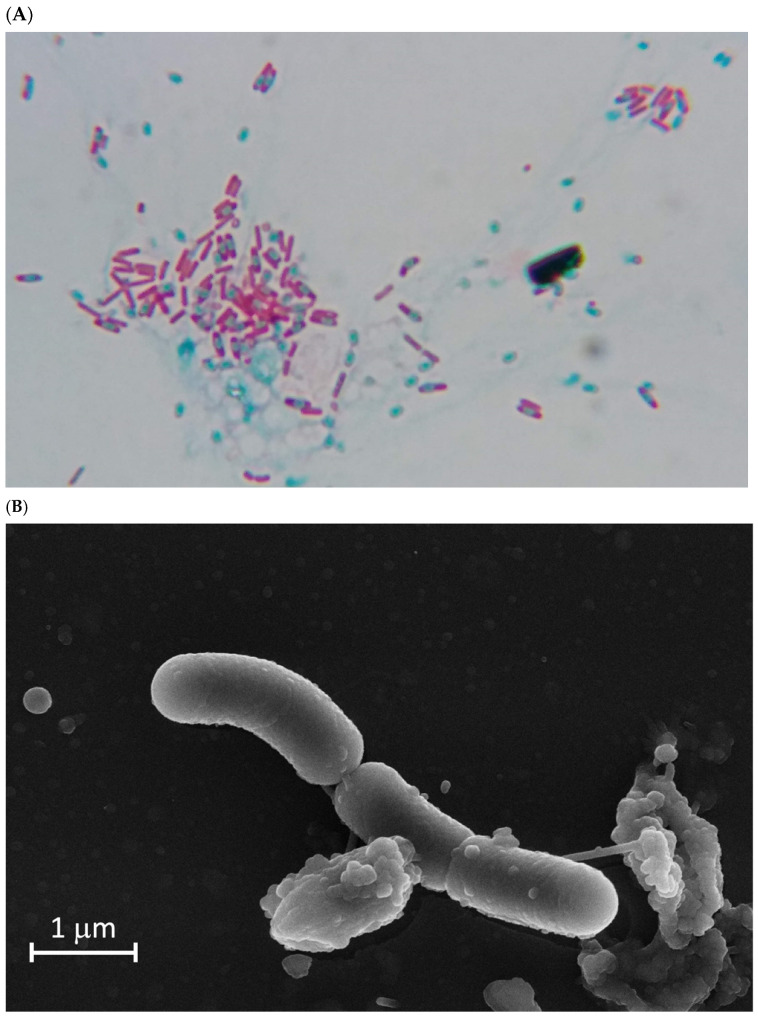
Optical, transmission, and scanning microscopy of the A6 isolate. Spore, endospore, and vegetative cell forms of isolate A6 cultured on TSB for 24 h. (**A**) Optical microscopy image, ×1000 magnification, where spores and endospores are stained with malachite green stain and appear light blue/green, while vegetative cells are stained with safranin and appear pink/red. (**B**) Scanning electron microscopy of vegetative cells of strain A6. (**C**) Transmission electron microscopy of spores of the A6 strain, showing the ultrastructure’s coat, corte, inner membrane, and core. Magnification is shown at the bottom of the electronic microscopy image.

**Figure 3 biology-13-00668-f003:**
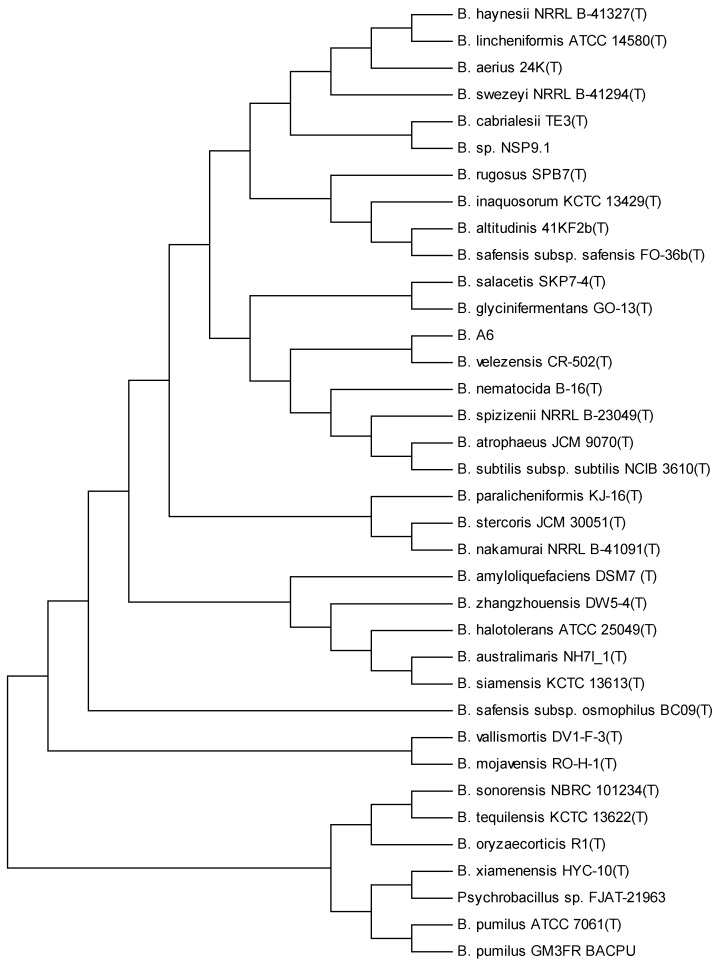
Taxonomic comparison of the A6 isolate with closely related *Bacillus* strains. Neighbour-joining phylogenetic tree based on 16S rRNA sequence (GenBank acc. no. SAMN40153660) comparisons of isolate A6 and its 35 closest relatives. Strains DSM7, D747, and FZB42 were used as the outgroups. The numbers at bifurcations indicate how many times each species coincided in this position as a percentage, and only values > 50% are shown. Bar, 0.01 changes per nucleotide position.

**Figure 4 biology-13-00668-f004:**
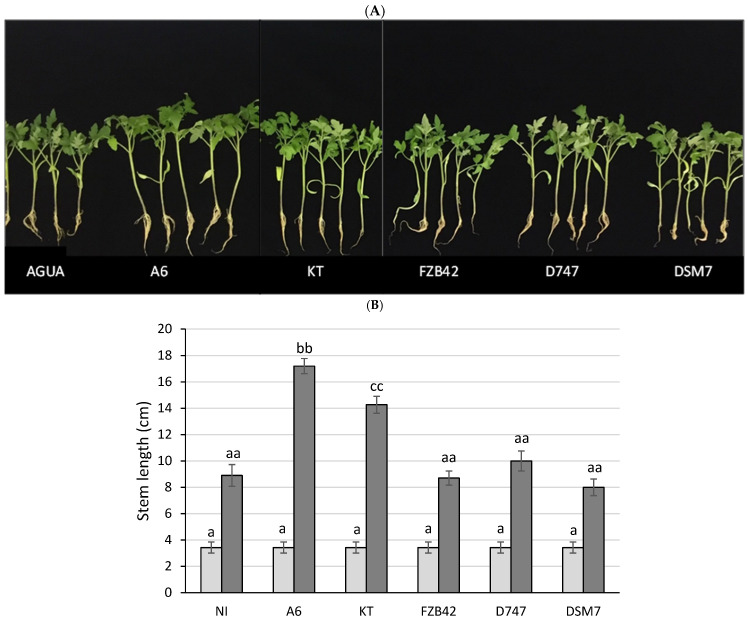
Tomato plants treated with different *Bacillus* strains under biostimulation conditions. Appearance of the plants inoculated with the different *Bacillus* strains at 31 days after inoculation. (**A**) Stem length (**B**), root length (**C**), and dry weight of the plants (**D**) are represented for each group of plants at 0 days (represented in light gray) and at 31 days (represented in dark gray) after inoculation. Significant differences are indicated by the presence of letters, with one letter corresponding to a period of 0 days and two letters corresponding to a period of 31 days. aa indicates that not there are differences with non-inoculated plants. bb and cc indicate higher dry weight values with respect to non-inoculated plants. dd indicates lower dry weight values than non-inoculated plants.

**Figure 5 biology-13-00668-f005:**
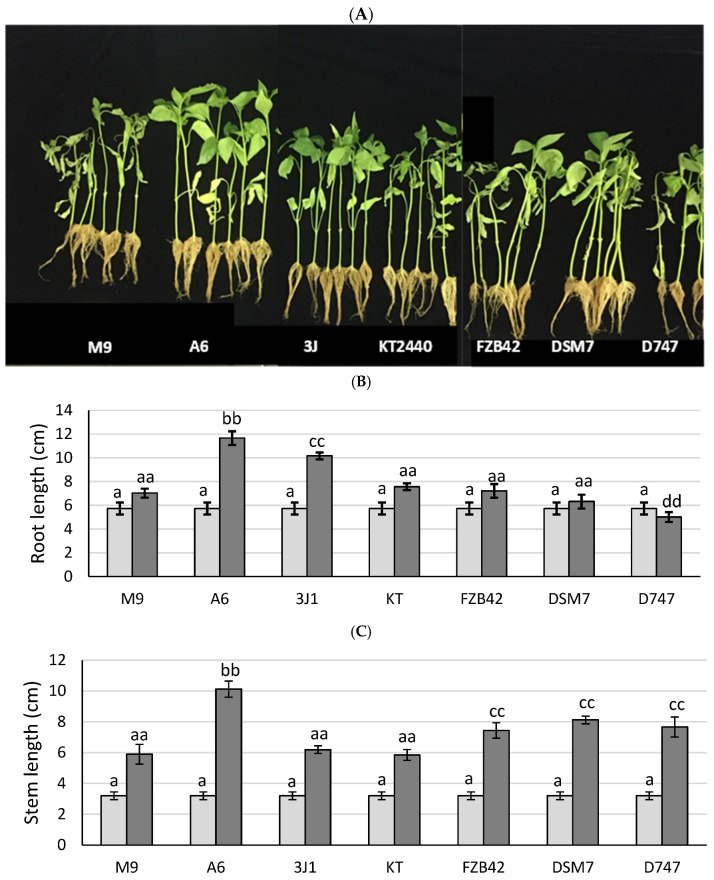
Pepper plants treated with different *Bacillus* strains under xeroprotection conditions. Appearance of the plants inoculated with the different *Bacillus* 32 days after sowing (**A**). Root lengths (**B**), stem lengths (**C**), fresh weights (**D**) and total dry weights (**E**) are represented for each group of plants at 0 days (represented in light gray) and at 31 days (represented in dark gray). Significant differences are indicated by the presence of letters, with one letter corresponding to a period of 0 days and two letters corresponding to a period of 32 days. aa indicates that not there are differences with non-inoculated plants. bb and cc indicate higher parameter values with respect to non-inoculated plants.

**Figure 6 biology-13-00668-f006:**
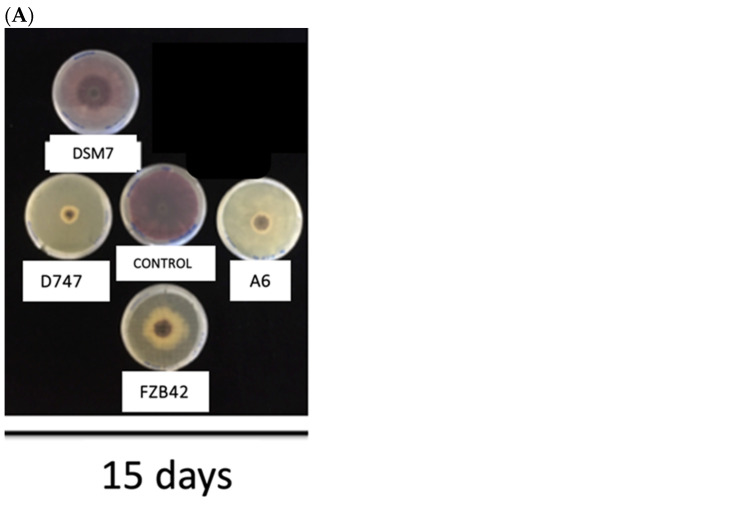
(**A**) Photograph of the antagonism test of *Bacillus* genus strains on *F. oxysporum*. The growth of the fungus *F. oxysporum* in the presence of the different strains after 15 days is shown. (**B**) Antagonism of *Bacillus* genus strains on *F. oxysporum.* The diameter growth of the *F. oxisporum* fungus is shown in the presence of the different *Bacillus* strains at day 5 (dark gray), day 10 (gray), and day 15 (light gray). The statistical tests that are represented with one letter correspond to the time of 5 days, those with two letters to the time of 10 days, and those with three letters to the time of 15 days. a, b and c indicate that the diameter of F. oxysporum is significantly smaller than that of the control.

**Figure 7 biology-13-00668-f007:**
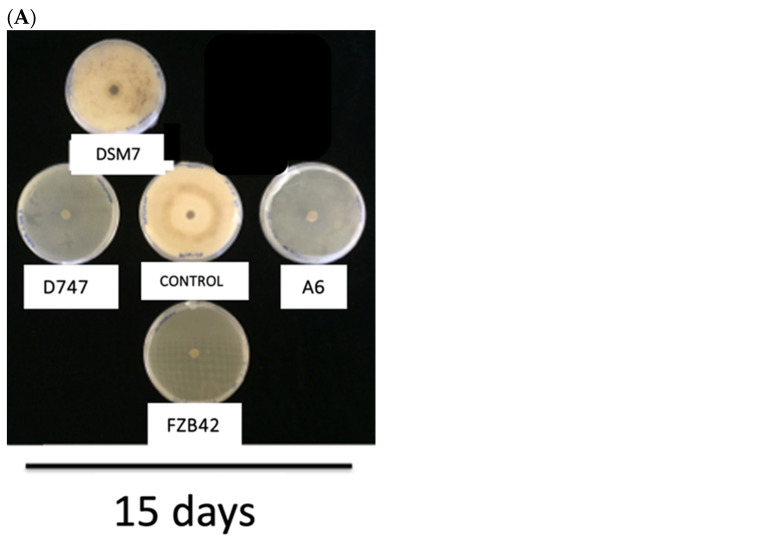
(**A**) Photograph of the antagonism test of *Bacillus* genus strains on *B. cinerea.* The growth of the fungus *B. cinerea* in the presence of the different strains after 15 days is shown. (**B**) Antagonism of *Bacillus* genus strains on *B. cinerea*. The diameter growth of the *B. cinerea* fungus is shown in the presence of the different *Bacillus* strains at day 5 (dark gray), day 10 (gray), and day 15 (light gray). The statistical tests that are represented with one letter correspond to the time of 5 days, those with two letters to the time of 10 days, and those with three letters to the time of 15 days. b and c indicate size significantly smaller than the control.

**Figure 8 biology-13-00668-f008:**
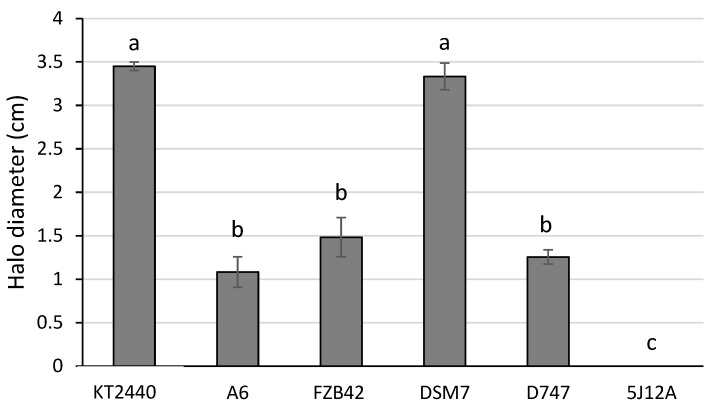
Diameters of halos produced by the solubilization of phosphates. Halo diameter values (cm) produced by the solubilization of phosphates of each strain. The *Arthrobacter koreensis* 5J12A strain was used as a negative control. a indicates that there are no differences in the halo diameter in the presence of KT2440. b and c indicate diameters significantly smaller than the KT2440 control.

**Figure 9 biology-13-00668-f009:**
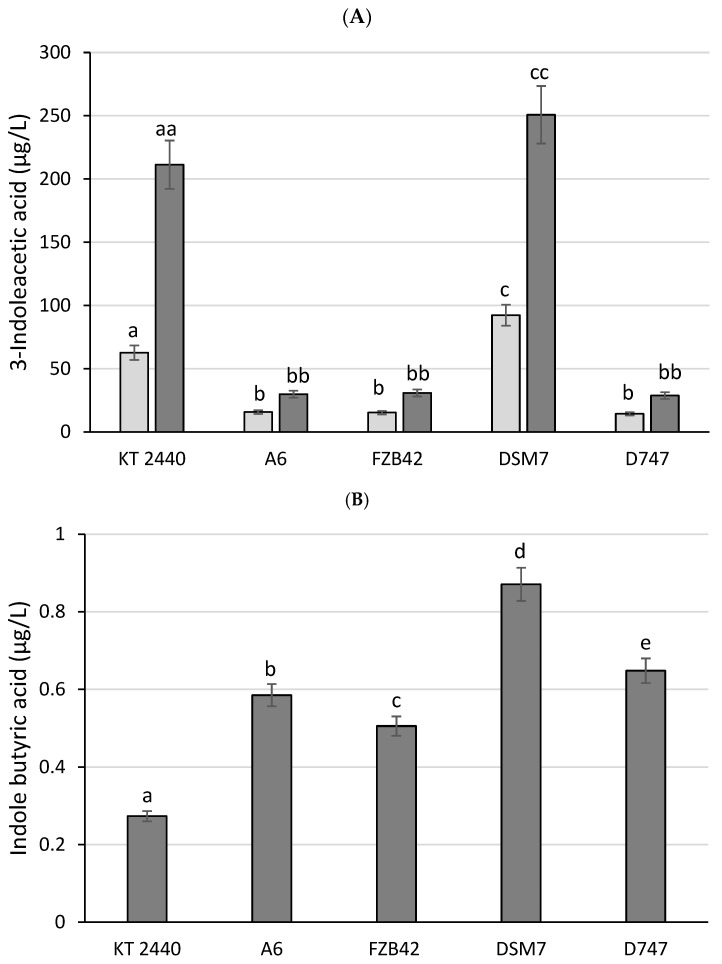
Production of 3-indoleacetic acid, indole butyric acid, and giberellic acid A3. Indole acetic acid production values for each strain are shown at 24 (light gray bars) and 72 h (dark gray bars) of incubation (**A**). Indole butyric acid production values for each strain at 72 h of incubation are depicted (**B**). Giberellic acid A3 production values for each strain at 24 h of incubation are also shown (**C**). a indicates no significant differences with the KT2440 control, while b, c, d, and e indicate significant differences with the KT2440 control.

**Table 1 biology-13-00668-t001:** Strains used in this work and their role in this study.

Strain	Role in This Study	Reference
*Pseudomonas putida* KT2440	PGPR Spore staining	[4]
*Escherichia coli* OP50	PGPR	[21]
*Microbacterium* sp. 3J1	PGPR	[4]
*Proteus* sp. S47	Siderophore negative control	[22]
*Bacillus amyloliquefaciens* D747	PGPR and fungicides	[15]
*Bacillus amyloliquefaciens* subsp. *amyloliquefaciens* Fukumoto DSM7	PGPR and fungicides	[13,14]
*Bacillus velezensis* FZB42	PGPR and fungicides	[16]
*Bacillus velezensis* A6	PGPR and fungicides	This study
*Bacillus subtilis* CECT39	Spore staining	[20]

**Table 2 biology-13-00668-t002:** The 36 strains most closely related to the A6 isolate based on digital DNA–DNA hybridization, accession numbers, and intergenomic distances expressed as % similarity are shown.

Number	Accession Number	Species	% Similarity
1	AJVF01000044→ 16S: MW578390	*B. siamensis* KCTC 13613(T)	99.93
2	AY603658	*B. velezensis* CR-502(T)	99.92
3	ABQL01000001→ 16S: MK559753	*B. subtilis* subsp. *subtilis* NCIB 3610(T)	99.78
4	FN597644→ 16S: AY055225	*B. amyloliquefaciens* DSM 7(T)	99.70
5	AY820954	*B. nematocida* B-16(T)	99.70
6	LSAZ01000028→ 16S: NR_151897	*B. nakamurai* NRRL B-41091(T)	99.63
7	AYTO01000043	*B. tequilensis* KCTC 13622(T)	99.55
8	MK462260	TE3(T)	99.55
9	AMXN01000021	*B.* KCTC 13429(T)	99.55
10	MN536904	*B. stercoris* JCM 30051(T)	99.55
11	JH600273→ NZ_JH600276	*B.* DV1-F-3(T)	99.48
12	JABUXO010000041→ 16S: MT554518	*B. rugosus* SPB7(T)	99.48
13	AB021181	*B. atrophaeus* JCM 9070(T)	99.40
14	LPVF01000003→ 16S: MN840041	*B. halotolerans* ATCC 25096(T)	99.40
15	CP002905→ 16S: NR_024931	*B. spizizenii* NRRL B-23049(T)	99.40
16	JH600280→ 16S: NZ_AYTL01000035	*B. mojavensis* RO-H-1(T)	99.33
17	LECW01000063	*B. glycinifermentans* GO-13(T)	98.28
18	KY694465	*B. paralicheniformis* KJ-16(T)	98.20
19	AUQZ01000032→16S: JF802181	*B.* sp. NSP9.1	98.05
20	AE017333→ 16S: LR594217	*B. licheniformis* ATCC 14580(T)	97.98
21	MRBL01000076→ NR_157609	*B. haynesii* NRRL B-41327(T)	97.90
22	AYTN01000016 *	*B. sonorensis* NBRC 101234(T)	97.75
23	MRBK01000096→ NR_157608	*B. swezeyi* NRRL B-41294(T)	97.75
24	AJ831843	*B. aerius* 24K(T)	97.37
25	ASJC01000029	*B. altitudinis* 41KF2b(T)	97.30
26	AMSH01000114	*B. xiamenensis* HYC-10(T)	97.22
27	LJIY01000004→ NZ_LJIY01000031	*Psychrobacillus sp*. FJAT-21963	97.22
28	ASJD01000027→ 16S: MK849613	*B. safensis subsp. safensis* FO-36b(T)	97.07
29	KY990920	*B. safensis subsp. osmophilus* BC09(T)	97.07
30	ABRX01000007→ 16S: MKZN01000032	*B. pumilus* ATCC 7061(T)	97.00
31	JOTP01000061→ 16S: MZ066819	*B. zhangzhouensis* DW5-4(T)	97.00
32	JX680098→ 16S: MW228046	*B. australimaris* NH7I_1(T)	96.92
33	MKZN01000032	*B. pumilus* GM3FR BACPU	96.77
34	LC367333	*B. salacetis* SKP7-4(T)	96.18
35	KF548480	*B. oryzaecorticis* R1(T)	96.17
36	ABCF01000001	*B.* sp. SG-1 1101501000768	96.10

**Table 3 biology-13-00668-t003:** Agricultural Protection Against Stress Index (APSI) tests on the OG*Ba* strains *B. velezensis* A6, *B. amyloliquefaciens* FZB42, *B. amyloliquefaciens* DSM7, and *B. amyloliquefaciens* subsp*. plantarum* D747. The maximum theoretical values are shown as well.

	Max Score	A6	FZB42	DSM7	D747
Root Length (PGPR)	5	5	3.75	3.75	3.75
Stem Length (PGPR)	5	5	5	3.75	2.5
Dry Weight (PGPR)	5	5	5	2.5	3.75
RWC (PGPR)	5	5	5	5	5
RWC (Drought)	9	9	6.75	6.75	6.75
Root Length (Drought)	9	9	6.75	6.75	4.5
Dry Weight (Drought)	9	9	6.75	6.75	6.75
Antagonism *B. cinerea*	7	7	7	1.75	7
Antagonism *F. oxysporum*	7	5.95	3.5	0	5.25
Antibiosis *B. cinerea*	7	3.5	3.5	1.75	3.5
Antibiosis *F. oxysporum*	7	3.5	1.75	1.75	3.5
Siderophores	4	4	4	4	4
Phosphate Solubilization	4	1	2	4	2
Potassium Solubilization	4	4	4	4	4
AIA Production	3	0.75	0.75	3	0.75
IBA Production	3	2.25	2.25	3	2.25
GB Production	3	0	0	3	0
Urea Hydrolysis	4	0		0	0
**Total Maximum Score**	100	78.95	67.75	61.5	65.25

## Data Availability

All data will be made available upon request.

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
