# Peer review of "Comparative Study of Bacillus-Based Plant Biofertilizers: A Proposed Index"

_biology, 2024, doi:10.3390/biology13090668_

Round 1
Reviewer 1 Report
Comments and Suggestions for Authors
The peer-reviewed article is the original work of the authors and is devoted to the topical and important topic of using biocontrol bacteria to protect agricultural plants from pathogenic fungi, including under drought conditions. The work was carried out under the conditions of laboratory cultivation of pepper and lettuce plants. The authors isolated A6 isolate, checked a number of indicators important from the point of view of agronomy both for this isolate and for known strains of Bacillus amyloliquefaciens and B. amyloliquefaciens subsp. plantarum, as well as the Agricultural Protection Against Stress Index determined the agronomic value of each of the studied strains. The Conclusions state that the isolated B. velezensis A6 strain has a high potential to be used as a plant biostimulant for plant protection under conditions of drought and the presence of phytopathogenic fungi. And it is also noted that the proposed Agricultural Protection Against Stress Index allows to compare the effectiveness of new and generally recognized strains.
I have a number of comments on the reviewed work:
1. It is absolutely necessary to register the obtained sequence of nucleotide sequences of a new isolate in the GenBank database.
2. The purpose of the research should be specified, because it is not specified.
3. The Materials and methods section should be rewritten so that this section becomes clear and understandable to the reader. It is important to indicate the references from which the research methods were taken. It is also not clear why this section lists lettuce plants in addition to pepper plants, and the Research Results section only contains pepper results.
4. Carefully approach the references used. There are a number of references that do not contain the information cited by the authors.
5. The values of the Agricultural Protection Against Stress Index in the text of the article and Table 3 differ. It is necessary to review the results again: where are the correct values - in the table or in the text?
6. Carefully correct all comments provided in the text of the article.
The article needs significant revision. It should be rewritten.

Author Response
Dear Reviewer,
First of all, thank you very much for your encouraging comments on our manuscript. We value your contributions in this regard to make a much better article from our manuscript.
We have tried to respond to all your comments, which we detail below:
COMMENT 1: “It is absolutely necessary to register the obtained sequence of nucleotide sequences of a new isolate in the GenBank database.”
RESPONSE 1: Thank you for the reminder. The nucleotide sequence of the new isolate is in the GenBank database now.
COMMENT 2: “The purpose of the research should be specified, because it is not specified.”
RESPONSE 2: Thanks again for pointing this requirement. We have included the purpose of the research, now is included in lines 114-118 of the revised manuscript.
COMMENT 3: “The Materials and methods section should be rewritten so that this section becomes clear and understandable to the reader. It is important to indicate the references from which the research methods were taken. It is also not clear why this section lists lettuce plants in addition to pepper plants, and the Research Results section only contains pepper results.”
RESPONSE 3: We agree with you in the fact that references were missing in addition to some text that was incorrectly allocated in the Results section when it was part of the Materials and Method section. We have rewritten the Material and method section, for a clearer version. Now this section includes the references from which the research methods were obtained. The reference to lettuce plants has been removed since this was part of experiments that were not included in the final version.
COMMENT 4: ”Carefully approach the references used. There are a number of references that do not contain the information cited by the authors.”
RESPONSE 4: The used references have been revised in order to show those that show relevant information. Thank you for pointing this mistake.
COMMENT 5: “The values of the Agricultural Protection Against Stress Index in the text of the article and Table 3 differ. It is necessary to review the results again: where are the correct values - in the table or in the text?”
RESPONSE 5: We appreciate this comment, as it has allowed us to include the right information in the text (corrections made in page 25).
COMMENT6: “Carefully correct all comments provided in the text of the article.”
RESPONSE 6: We are very grateful for your effort to make a much better article. We have gone through all your comments included in the PDF version that we think are very constructive. Once again, thank you very much for your help.
We very much appreciate your comments and think they have helped us in making a much better article.
Sincerely,
Maximino Manzanera.

Reviewer 2 Report
Comments and Suggestions for Authors
The manuscript entitled “Plant growth-promoting fungicide and drought-protective effect of strains of the genus Bacillus: Proposed Agricultural Protection Against Stress Index (APSI) protocol” is well-designed and has novel and interesting results. However, there are some items that should be considered in the manuscript as follows:
- The abstract comprehensively covers the study's objectives, methods, and findings. However, it could be more concise. Consider defining "OGBa" in the abstract to avoid confusion.
- The mention of the Delphi method and Probit function is appropriate but might benefit from a brief explanation of their significance in this context.
- Refine for conciseness, for instance, "Such an offer makes it difficult" could be "This abundance makes it challenging."
- The introduction provides a thorough background, covering the historical context and current challenges in agriculture, which is beneficial for understanding the study's relevance. Break down the introduction into smaller paragraphs for better readability and logical flow. Ensure each paragraph focuses on a single main idea to maintain coherence. Summarize some of the more detailed points or move them to a literature review section if appropriate.
- Clearly define acronyms and terms when first introduced, such as "CAP" and "PGPR."
- The introduction is dense with information, which is valuable but can be overwhelming. The transition from the general context to the specific focus of the study could be smoother.
Materials and Methods
- Specify the exact sources of media components and any modifications made to standard protocols, especially if deviations are significant for your study.
- Consider including details on how purity and identity of cultures were confirmed, which is crucial for reliability.
- Provide more specifics on the rationale behind using chloroform treatment for soil samples and its potential impact on microbial communities.
- Clarify the criteria used for selecting colonies from TSA plates for further studies.
- Mention the specific growth conditions used for the 3-day-old colonies before and after incubation at 72°C to ensure reproducibility.
- Include more details on the exact staining protocols used for spore staining and the duration of incubation for different stages (96 and 120 hours).
- Specify the exact parameters measured for pepper and lettuce plants (e.g., units of measurement for dry weight, fresh weight, etc.).
- Clarify the methodology used to ensure consistency in measurements across different time points.
- Provide more details on the rationale for choosing specific inoculum concentrations and the method used to determine these concentrations accurately.
- While you mentioned using RStudio for statistical analysis, consider specifying the exact tests used (e.g., ANOVA, Tukey's HSD) and providing more details on how data normality and outliers were assessed.
- Provide more specifics on the bioinformatics tools and pipelines used for genome assembly, annotation, and taxonomic analysis.
- Include relevant metrics (e.g., assembly quality metrics, ANI, dDDH) used to validate genome sequences.
Results
- The section is well-structured, with subsections clearly delineated for different experiments and analyses. This organization helps readers follow your experimental workflow and understand the logical progression of your findings.
- The method of isolating drought-protecting spore-forming strains from soil samples using chloroform treatment and subsequent culturing is well-described. Mentioning the minor modifications made to the established methods adds transparency to your procedures.
- Clear descriptions of the experimental setup for evaluating A6's ability to promote plant growth and protect against drought stress (e.g., using pepper plants) help in replicating your experiments.
Comments on the Quality of English LanguageThere are some typos in the whole text. It should be edited and polished
Author Response
RESPONSE TO REVIEWER 2
Subject: Response to Reviewer's Comments on Our Manuscript
Dear Reviewer,
We would like to express our appreciation for your encouraging feedback on our manuscript. Your insights and suggestions have been invaluable in guiding us to enhance the quality of our work.
In response to your comments, we have made considerable efforts to address each point raised. Below, we provide a detailed account of our responses:
Thank you once again for your thoughtful review.
COMMENT 1: “The abstract comprehensively covers the study's objectives, methods, and findings. However, it could be more concise. Consider defining "OGBa" in the abstract to avoid confusion.”
RESPONSE 1: Thank you for pointing in this respect. We have rewrite a more concise abstract and defined OGBa to avoid confusion, removing all abbreviations from the abstract.
COMMENT 2: “The mention of the Delphi method and Probit function is appropriate but might benefit from a brief explanation of their significance in this context.”
RESPONSE 2: Thanks again for making clear the need of better definition in this respect. We have changed the text from the initial lines 28-31 to the current lines 33-39 for a better understanding of Delphi method and Probit function.
COMMENT 3: “Refine for conciseness, for instance, "Such an offer makes it difficult" could be "This abundance makes it challenging.".”
RESPONSE 3: We agree with you in the fact that some statements needed to be refine for conciseness, therefore we have changed former line 21 to current line 26.
COMMENT 4: ”The introduction provides a thorough background, covering the historical context and current challenges in agriculture, which is beneficial for understanding the study's relevance. Break down the introduction into smaller paragraphs for better readability and logical flow. Ensure each paragraph focuses on a single main idea to maintain coherence. Summarize some of the more detailed points or move them to a literature review section if appropriate.”
RESPONSE 4: We appreciate very much your comment in this respect. We have breaken down the introduction into smaller paragraphs and improved the logical flow. We hace ensured that each paragraph focuses on a single idea and consequently we have improved coherence. We have summarized some of the more detailed points, hopefully to your satisfaction.
COMMENT 5: “Clearly define acronyms and terms when first introduced, such as "CAP" and "PGPR.”
RESPONSE 5: Once again, we appreciate this comment, we have defined acronyms and terms when first introduced. See lines 60 or 89 for examples.
COMMENT 6: “The introduction is dense with information, which is valuable but can be overwhelming. The transition from the general context to the specific focus of the study could be smoother.”
RESPONSE 6: We completely agree with you in this respect. We have produced a new introduction reducing the information smoothing the general context to the specific focus of the study. See lines 45-110.
COMMENT 7: For Materials and Method section “Specify the exact sources of media components and any modifications made to standard protocols, especially if deviations are significant for your study.”
RESPONSE 7: We have included references to the standard protocols as exact sources of materials and methods. Thank you.
COMMENT 8: For Materials and Method section “Consider including details on how purity and identity of cultures were confirmed, which is crucial for reliability.”
RESPONSE 8: Again, we appreciate your suggestion. We have given details on how purity and identity of cultures were confirmed. Lines 137-148. “After 48 h of incubation at 30°C, single colonies were selected from the agar plates and re-streaked onto fresh selective media to obtain pure cultures. To select for different isolates, morphology (size, shape, color, and texture of colony) and growth characteristic (rate of growth, colony edge and elevation) were followed. This step was repeated at least three times to ensure the elimination of contaminants. To confirm the purity of the cultures, individual colonies from each re-streaked plate were subjected to microscopic examination. A sterile slide and coverslip were used to observe colony morphology and cellular characteristics under [e.g., 100x] using a light microscope. The absence of contaminants was verified by assessing the homogeneity of cell morphology. When required, molecular identification was performed, by DNA extraction from purified cultures using the DNeasy® Blood and Tissue Kit (Qiagen, Venlo, The Netherlands), following the manufacturer’s instructions. The quantity and quality of the extracted DNA was performed using a NanoDrop 2000 (Thermo Scientific, Wilmington, CA, USA). Polymerase chain reaction (PCR) was performed with specific primers targeting the 16S rRNA gene to amplify a region characteristic of the genus and specie as previously described. PCR products were analyzed via agarose gel electrophoresis and sequenced as described below.”
COMMENT 9: “Provide more specifics on the rationale behind using chloroform treatment for soil samples and its potential impact on microbial communities.”
RESPONSE 9: We have provided specifics on the rationale behind using chloroform treatment for soil samples. See lines 127-132. Thank you for stating this important flaw. “To isolate spore forming microorganisms, 1 g of rhizospheric soil samples were air-dried and treated with pure chloroform (99.8%) for 60 min with occasional vortexing as described in [13] since this process is detrimental for most vegetative cells, while many spores resist the treatment with organic solvents.”
COMMENT 10: “Clarify the criteria used for selecting colonies from TSA plates for further studies.”
RESPONSE 10: We understand that there was a lack of explanation for the criteria used for selecting colonies. This has been solved with lines 133-136
COMMENT 11: “Mention the specific growth conditions used for the 3-day-old colonies before and after incubation at 72°C to ensure reproducibility.”
RESPONSE 11: We have given the specific growth conditions used for these colonies (see lines 157-161). “Aliquot volumes (10 μL) were plated in TSA plates before and after the 72°C incubation, and incubated again at 30°C for 3 days. Strains able to grow in TSA plates at 30°C under both conditions were selected as spore-producing or thermotolerant bacteria”
COMMENT 12: “Include more details on the exact staining protocols used for spore staining and the duration of incubation for different stages (96 and 120 hours).”
RESPONSE 12: The required details can be found in lines 164-166 “For spore staining 100 mL flasks containing 10 mL of TSB were inoculated with the different candidates, and incubated at 30°C with shaking at 180 rpm (Infors HT Multitron).”
COMMENT 13: “Specify the exact parameters measured for pepper and lettuce plants (e.g., units of measurement for dry weight, fresh weight, etc.).”
RESPONSE 13: Thank you for stating this important point. The exact parameters measured for plants have been included in lines 197-201. “For Italian sweet pepper plants (Capsicum annuum L. cv. Maor), dry weight (DW), fresh weight (FW), fully turgid weight (FTW) of the whole plants free from soil were measured four times after inoculation at 0 and 31 days after inoculation and expressed in grams. The relative water content (RWC) was calculated according to [13,20] as follows: RWC = (FW-DW) × (TW-DW)-1. The relative water content is a dimensionless parameter whose maximum theoretical value is 1 in a state of maximum hydration”
COMMENT 14: “Clarify the methodology used to ensure consistency in measurements across different time points.”
RESPONSE 14: We have clarified the methodology used to ensure consistency in measurements across different time points (lines 202-210). “To ensure consistency in measurements across different time points to determine fresh weight, plants were harvested at the same time of day (10:00 am) to avoid variations due to transpiration. A precision scale was used to record weight immediately after harvest. To obtain the fully turgid weight, the samples were immersed in distilled water for 48 hours in the dark. After this period, they were removed, gently dried with absorbent paper, and weighed again. As for the dry weight measurement, samples were placed in an oven at 70°C for 72 hours to ensure complete removal of moisture. Once cooled in a dehydrator, they were weighed to obtain the dry weight.
COMMENT 15:” Provide more details on the rationale for choosing specific inoculum concentrations and the method used to determine these concentrations accurately.”
RESPONSE 15: The rationale for choosing specific inoculum concentrations and the method used to determine these concentrations are now in lines 218-226. “. For the inoculation with the different strains inocula, and to guarantee a similar number of bacterial cells in each plant, and that their effect came from the plant-microorganism interaction, without taking into account the presence of secondary metabolites produced throughout the culture, the cells were separated from the culture by centrifugation and resuspended in M9 buffer. These inocula were supplied by VitaNtech Biotechnology, Spain and plants were treated with 3 mL of the liquid inoculant (consisting of a bacterial suspension from the enriched cultured on M9 buffer (Na2HPO4·7H2O; KH2PO4; NH4Cl; NaCl) at an absorbance of 1600 nm) representing a concentration between 1·106 and 1·108 of colony-forming units (CFU).”
COMMENT 16: “While you mentioned using RStudio for statistical analysis, consider specifying the exact tests used (e.g., ANOVA, Tukey's HSD) and providing more details on how data normality and outliers were assessed.”
RESPONSE 16: We agree that i important to give details on how data normality and outliers were assessed. Therefore, we have modified the text corresponding to subsection 2.15. like this “For the statistical treatment of the data, RStudio i386 software, version 4.0.3 (PBC, Boston, MA, USA, 2011) was used. An evaluation of significant differences between treatments was carried out by applying a 95% confidence interval. The data were analyzed using an ANOVA (analysis of variance) model to determine whether there were statistically significant differences between the means of the compared groups. If significant differences were found, a post hoc analysis was conducted using Tukey's HSD (Honest Significant Difference) procedure to compare the means of all groups exhaustively, allowing for pairwise comparisons.
To assess the normality of the data, a Shapiro-Wilk test was performed, and it was considered that the data were normally distributed if the resulting p-value was greater than 0.05. Additionally, the Bonferroni outlier test was employed to identify potential outliers. This procedure was carried out by checking whether any observations deviated significantly from the evaluated normal distribution. If any significant deviation was noted, the corresponding data were excluded from the analysis to ensure the robustness of the results.
It was considered that the difference between treatments was statistically significant when the obtained p-value was less than 0.05, allowing us to reject the null hypothesis of equality of means.”
COMMENT 17: “Include relevant metrics (e.g., assembly quality metrics, ANI, dDDH) used to validate genome sequences.”
RESPONSE 17: We have included the followin information in line 315 that hope would satisfy you “Reads were assembled into contigs by SPAdes v3.13.0 [25], using 55, 75 and 97 as k-mer sizes.”
COMMENT 18: “The method of isolating drought-protecting spore-forming strains from soil samples using chloroform treatment and subsequent culturing is well-described. Mentioning the minor modifications made to the established methods adds transparency to your procedures.”
RESPONSE 18: Thank you for your encouraging comments on the Results section. With respect to this specific point, we have modified the text in this sense in the Material and Method section, as above indicated for comment 9.
COMMENT 19: “Clear descriptions of the experimental setup for evaluating A6's ability to promote plant growth and protect against drought stress (e.g., using pepper plants) help in replicating your experiments.”
RESPONSE 19: Once again, thank you for helping us in creating a much better manuscript with your comments. We have modified the text in lines 446-450 with the following information “While strains that produced an increase in dry weight, as well as relative water content in the absence of irrigation above that observed in non-inoculated plants, were considered protective against drought.”
COMMENT 20: “There are some typos in the whole text. It should be edited and polished”
RESPONSE 20: Hopefully we have edited and polished the text to remove typos.
We very much appreciate your comments and think they have helped us in making a much better article.
Sincerely,
Maximino Manzanera.

Reviewer 3 Report
Comments and Suggestions for Authors
The authors of the original study isolated and tested a strain of a potentially beneficial bacterium for cometabolism with plants in drought conditions. A wide variety of necessary methods and tests indicate the good significance of their results. It is extremely rare to find a phosphate solubilization test in such articles. The authors understand its importance in studying the effect of increasing the growth rate of plants and made this test. But I have a few questions. Paragraph 2.2. Isolation of drought-resistant bacterial strains: it is not entirely clear why the chloroform procedure was performed during isolation? Is it for the extraction of metagenomic DNA? Then it needs to be moved to another point. There is a link in the text to the previous publication of the authors, but I think that in this article it is necessary to briefly describe the meaning of this method. Fig. 2 There is no declared and described transmission microscopy I liked the scale of values the Agricultural Protection Against Stress Index proposed by the authors. It would be good to expand abitotic stress with a few more points. The authors mentioned tests on plants in the fields – it is necessary to describe how the plants were treated with microbial suspension and whether the presence of their bacteria in the studied samples of lettuce and corn was monitored. The methods only use laboratory processing. Or even add field research to the main results and describe everything in detail. Perhaps it will be in the next article, but is it necessary now in CONCLUSIONS? The article deserves high praise and can be published after minor edits.
Author Response
RESPONSE TO REVIEWER 3
Dear Reviewer,
Thank you sincerely for your positive feedback on our manuscript. We greatly appreciate your insights, which have helped us enhance the quality of our article. We have made an effort to address all your comments, and we have outlined our responses below:
COMMENT 1: “Paragraph 2.2. Isolation of drought-resistant bacterial strains: it is not entirely clear why the chloroform procedure was performed during isolation? Is it for the extraction of metagenomic DNA? Then it needs to be moved to another point. There is a link in the text to the previous publication of the authors, but I think that in this article it is necessary to briefly describe the meaning of this method.”
RESPONSE 1: Thank you for pointing this lack of clarity. We have modified the text in paragraph 2.2. by adding the following text “To isolate spore forming microorganisms, 1 g of rhizospheric soil samples were air-dried and treated with pure chloroform (99.8%) for 60 min with occasional vortexing as described in [13], since this process is detrimental for most vegetative cells, while many spores resist the treatment with organic solvents.” That we hope it will help the reader in understanding the rationale in using chloroform.
COMMENT 2: “Fig. 2 There is no declared and described transmission microscopy.”
RESPONSE 2: Thanks again for pointing this mistake. By some sort of error, the transmission microscopy was missing. It is included now.
COMMENT 3: “The Materials and methods section should be rewritten so that this section becomes clear and understandable to the reader. It is important to indicate the references from which the research methods were taken. It is also not clear why this section lists lettuce plants in addition to pepper plants, and the Research Results section only contains pepper results.”
RESPONSE 3: We agree with you in the fact that references were missing in addition to some text that was incorrectly allocated in the Results section when it was part of the Materials and Method section. We have rewritten the Material and method section, for a clearer version. Now this section includes the references from which the research methods were obtained. The reference to lettuce plants has been removed since this was part of experiments that were not included in the final version.
COMMENT 4: ”Carefully approach the references used. There are a number of references that do not contain the information cited by the authors.”
RESPONSE 4: The used references have been revised in order to show those that show relevant information. Thank you for pointing this mistake.
COMMENT 5: “The values of the Agricultural Protection Against Stress Index in the text of the article and Table 3 differ. It is necessary to review the results again: where are the correct values - in the table or in the text?”
RESPONSE 5: We appreciate this comment, as it has allowed us to include the right information in the text (corrections made in page 25).
COMMENT6: “Carefully correct all comments provided in the text of the article.”
RESPONSE 6: We are very grateful for your effort to make a much better article. We have gone through all your comments included in the PDF version that we think are very constructive. Once again, thank you very much for your help.
We very much appreciate your comments and think they have helped us in making a much better article.
Sincerely,
Maximino Manzanera.

Reviewer 4 Report
Comments and Suggestions for Authors
Dear Authors
I have thoroughly reviewed your manuscript entitled: "Plant growth-promoting fungicide and drought-protective efect of strains of the genus Bacillus: Proposed Agricultural Protection Against Stress Index (APSI) protocol".
The topic presented is novel and of great importance to the scientific community. I have made some observations, I recommend you consider them for the improvement of your manuscript.
Introduction
L48-49: 7/16/2024?? I don't understand this text. Check
L50-54: No paragraph can be left without biblical quotations. Everything that is written must be justified with scientific documents.
L80: Check and correct: "And"
L80: These words should be capitalised first letter: plant growth-promoting rhizobacteria
Results
L559: Correct the table, check the rules provided by the journal to the authors.
L367: Please use italics for scientific names. Revise this throughout the manuscript
DISCUSION
The discussion should expand and use abundant scientific literature to analyse the results obtained vs. previous studies.
CONCLUSIONS
Further conclusions are to be extended taking into consideration the relevance of the res
ults obtained.
Author Response
RESPONSE TO REVIEWER 4
Dear Reviewer,
We greatly appreciate your positive feedback on our manuscript. Your insightful comments have played a crucial role in enhancing the quality of our article. We have thoroughly addressed each of your suggestions and outlined our responses below.
COMMENT 1: “
L48-49: 7/16/2024?? I don't understand this text. Check.”
RESPONSE 1: Thank you for pointing this mistake. We have modified the text that appears as follows: “Recent reviews highlight the significance of these approaches in mitigating the detrimental impact of conventional agriculture practices [4].”
COMMENT 2: “L50-54: No paragraph can be left without biblical quotations. Everything that is written must be justified with scientific documents.”
RESPONSE 2: Thanks again for pointing this mistake. We have included biblical quotations as pointed. “Furthermore, the new European Common Agrarian Policy (CAP) aims to achieve a 50% reduction in the use of high-risk phytosanitary products (i.e. insecticides, fungicides, herbicides, rodenticides and nematicides of chemical origin) and a 50% decrease in soil nutrient losses by 2030. Notably, this policy seeks to limit the use of conventional fertilizers by 20% while promoting the increased adoption of biostimulants and biofertilizers as essential alternatives. An additional factor driving these changes is the rising consumer demand for organic foods, which has surpassed 106 billion euros globally. Within the European market alone, this expenditure is 46 billion euros, with a future projection that organic production will occupy 25% of available farmland [5].”
COMMENT 3: “L80: Check and correct: "And"”
RESPONSE 3: We agree with you in the fact that this should be corrected. We have produced a new Intruduction as suggested by one of the other Reviewers. This new Introduction, corrected your suggestion as well.
COMMENT 4: “L80: These words should be capitalised first letter: plant growth-promoting rhizobacteria”
RESPONSE 4: We have capitalized the words as suggested “Given the diverse range of bacteria functioning as Plant Growth-Promoting Rhizobacteria (PGPR), a standardized system for comparing these microorganisms is necessary for the benefit of end users.
COMMENT 5: “L559: Correct the table, check the rules provided by the journal to the authors.”
RESPONSE 5: Thank you for your observation. The table has been corrected following the journal´s instructions for authors.
COMMENT 6: “L367: Please use italics for scientific names. Revise this throughout the manuscript”
RESPONSE: Scientific names have been revised and now appear in italic. Thank you for your indication.
COMMENT 7: “The discussion should expand and use abundant scientific literature to analyse the results obtained vs. previous studies.”
RESPONSE 7: Discussion has been expanded and references to scientific literature supporting the results and previous studies has been included.
COMMENT 8: “Further conclusions are to be extended taking into consideration the relevance of the results obtained.”
RESPONSE 8: Thank you for your supporting statements. Further conclusions are included now.
We are extremely thankful for your comments, as they have been essential in guiding us to enhance our article.
Sincerely,
Maximino Manzanera.

Round 2
Reviewer 1 Report
Comments and Suggestions for Authors
Most of the comments have been corrected by the authors. However, some mistakes should be corrected:
1) Line 120 - Please sign the second column
2) Lines 148, 421, Table 2 – “species” is written in the singular and plural
3) Lines 291, 299 - not the oven, but the thermostat
4) Lines 383-385 - Please place the accession number after the number of base pairs: ...(1,430 bp) (Acc. No. PQ099275), which showed...
After correcting minor comments, the article can be published

Author Response
Most of the comments have been corrected by the authors. However, some mistakes should be corrected:
- Line 120 - Please sign the second column
Answer: We thank the reviewer for the comment. Please the revised version of table 1
- Lines 148, 421, Table 2 – “species” is written in the singular and plural
Answer: It has been corrected in lines 148, 421 and table 2
- Lines 291, 299 - not the oven, but the thermostat
Answer: We thank the reviewer for the comment. It has been corrected in lines 291 and 299
- Lines 383-385 - Please place the accession number after the number of base pairs: ... (1,430 bp) (Acc. No. PQ099275), which showed...
Answer: Please see the new phrase in lines 383-385:” (Acc. No. PQ099275)”
